# Nutritional Characterization of Hay Produced in Campania Region: Analysis by the near Infrared Spectroscopy (NIRS) Technology

**DOI:** 10.3390/ani12213035

**Published:** 2022-11-04

**Authors:** Fabio Zicarelli, Fiorella Sarubbi, Piera Iommelli, Micaela Grossi, Daria Lotito, Pietro Lombardi, Raffaella Tudisco, Federico Infascelli, Nadia Musco

**Affiliations:** 1Department of Veterinary Medicine and Animal Production, University of Napoli Federico II, 80100 Napoli, Italy; 2Institute for the Animal Production System Mediterranean Environment, National Research Council, 80055 Portici, Italy

**Keywords:** hays, NIRS (Near Infrared Spectroscopy), ruminants’ feeding

## Abstract

**Simple Summary:**

Forage is the basis of ruminants’ diet and its nutritional quality strongly influences the productive performance and the health status of animals. Analyzing forage characteristics in a timely and accurate way is, therefore, critical to formulate an adequate ration that can meet the animal needs. For this purpose, the use of the Near Infrared Spectroscopy (NIRS) technology is an excellent alternative method to traditional laboratory analysis. In this trial, more than 400 hay samples collected in the Campania region (southern Italy) were analyzed by using the NIRS technology. Alfalfa hay produced in the Piana del Sele area seems the most promising hay, characterized by high protein levels and low structural carbohydrates. On the contrary, the polyphite and Gramineae hays produced in most of the areas of Campania region showed poor nutritional value due to the low protein content and high structural carbohydrate that significantly reduced its digestibility. Results showed that hay quality strongly differs among different areas, and confirmed the usefulness of the NIRS technology to obtain a fast and accurate analysis of forage quality, thus allowing the nutritionists to formulate adequate animal rations and to reveal and correct a number of management factors in order to achieve the highest feed quality to obtain the best performance ensuring animal health.

**Abstract:**

Since the dietary characteristics of hays can be very variable, it is of great importance for nutritionists to know their chemical composition in order to formulate adequate rations for the animals. Laboratory analyses are time-consuming and expensive while the Near Infrared Spectroscopy offers several advantages, including obtaining information on feeds nutritional characteristics very quickly and in situ at the farm, thanks to portable NIRS. In this trial, over 400 hay samples collected in the Campania region (Italy) were analyzed with portable NIRS device. The final aim was to analyze the differences in hay quality in different production areas, highlighting the possible factors involved and suggesting possible corrective measures. All the analyzed hays (polyphite, Gramineae and alfalfa) were significantly (*p* < 0.05) influenced by the area of cultivation/preservation. In particular, the polyphite and Gramineae hays produced in most of the areas of Campania region showed poor nutritional value due to the low protein content and high structural carbohydrate that significantly reduced its digestibility. The use of high-quality forages in the ration represents a pivotal factor to allow the production of high-quality products of animal origin. The use of NIRS seems to be a valuable strategy to select potential treatments that can increase feed digestibility and to avoid long chemical analysis.

## 1. Introduction

Haymaking is one of the oldest methods used to preserve forage by reducing moisture content until reaching 15–20% of dry matter; thus, hay is essentially biologically inactive respect to both plant enzyme activity and microbial spoilage [1,2].

Several factors (i.e., type of forage, soil fertilization, plant vegetative stage at cut, cutting method, climatic conditions, overturn and harvesting the forage mass and, finally, the characteristics of the storage site) can affect hay final quality [3,4].

When compared to fresh forage, hay’s nutritive value is lower, but if the different production phases are performed carefully, the losses in nutrient can be limited [5]. High quality hay in ruminant ration can improve the rumen fermentation characteristics [6,7] resulting in positive effects on animal well-being [8,9], as well as in a high nutritional quality of food (i.e., meat, milk, dairy products) [10,11].

In Italy, haymaking is the most used system for forage preservation. In particular, in Campania region the hays are mainly produced in the hilly areas of Avellino, Benevento and Salerno and in the plain areas of Salerno provinces either in extensive livestock to obtain high quality meat (i.e., Vitellone bianco dell’Appennino Centrale IGP (http://www.vitellonebianco.it (accessed on 27 October 2022)) and milk and dairy products (i.e., Latte Nobile^®^, La Compagnia della Qualità SRL, Napoli, Italy, http://www.lattenobile.it (accessed on 27 October 2022)) or intensive livestock to produce the Mozzarella di Bufala DOP, Campania, Italy, (http://www.mozzarelladop.it (accessed on 27 October 2022)) and buffalo meat [12].

Knowing the nutritive value of hay is of great importance for formulating balanced diets able to maintain animal health and to guarantee the high quality level of the products of animal origin. In addition, the use of locally produced hay has economic advantages for the breeder [13], as well as environmental rewards in terms of global warming potential as reported in a Life Cycle Assessment (LCA) study [14].

In recent years, Precision Livestock Farming (PLF) has significantly developed. Cattle breeding is one of the most affected by PLF, since it allows to obtain a greater profitability with the improvement of animal welfare. As reported by Buller et al. [15], PLF technologies have significant potential to increase monitoring and addressing animal welfare. They allow to monitor farmed animal health and welfare enabling a both fast and accurate treatment of diseases and or behavioral disorders as well as the prevention of their spread to other animals. Due to the close relationship between the ration composition and health in farm animals, it is a consequence that some PLF techniques, such as on field NIRS, may be greatly useful in improving animal welfare by a quick evaluation of feed quality and a timely correction of dietary imbalances. As a consequence, the accurate knowledge of feed composition should reduce feeding disorders and diet related diseases. The main technological innovations have to take into account all aspects related to farming, that can be monitored through sensors designed to acquire the raw data of interest, which have to be managed and stored to be accessible: the challenge is to obtain the largest amount of data automatically, quickly and accurately, aiming to increase automatic, precise, and accurate farm management. An important part of PLF is Precision Feeding (PF).

The control and a more detailed knowledge of feeding is a very essential aspect of breeding, due to the impact of nutrition on the farm balance sheet. On the other hand, the costs and the time to obtain accurate feed analysis is often a critical aspect. The Near Infrared Spectroscopy (NIRS) can be used on the farm in a PF system. The use of NIRS is a valuable strategy for selecting potential treatments that may increase feed digestibility and for avoiding time-consuming chemical analysis [16]. NIRS analysis offers the promise of a rapid, low-cost analysis of nutrient composition that could be applied to the increasing need for efficiency in livestock feeding. The time required for a single run can range from seconds to minutes. This method makes it possible to quickly study many samples with considerable time and cost savings compared to traditional techniques. Moreover, NIRS portable tools can be used on the barn directly, allowing a timely intervention of prevention and/or correction, as well as carrying out countless self-control analyses. However, the limitations to the spread of these technologies in the contest of commercial farms include the investment cost and the difficult amortization for the purchase of technologies, especially in small and medium farms.

For these reasons, in this study, we characterized forages preserved as hay produced in different areas of Campania region (Italy) by using the Near Infrared Spectroscopy (NIRS) technology. The final aim was to the analyze the differences in hay quality in different areas of production highlighting possible factors involved and suggesting possible correctives.

## 2. Materials and Methods

### 2.1. Samples Collection and Preparation

The hays production on the farm is a very common practice in Campania region and, since hay represents a very high percentage of the ruminants’ ration, it is important to know their nutritional characteristics in order to improve the production system and/or formulate suitable rations that take into account the other feeds to be supplemented in order to make up for the shortcomings of the hays themselves.

During the experimental period (July 2020 to December 2021) a total of 438 hay samples—127 polyphite, 117 Gramineae and 194 alfalfa—were collected. All the samples were farm-produced hays; thus, the forages were grown on the land located in six areas of Campania region (Southern Italy) (Figure 1):Avellino - hilly area;Benevento - plain area;Caserta - plain area;Salerno - plain area;Piana del Sele - plain area of Salerno province;Vallo di Diano - hilly area of Salerno province.

For the study, we have considered the hay production in the farms located in the Campania region, and not all the hays (polyphite, Gramineae and alfalfa) were produced in all the experimental areas investigated for this research. For hays production, all the legume forages were cut at the beginning of flowering, while those of Gramineae were cut at the beginning of earing, the time was chosen according to the environmental characteristics of each area. For each sampling site, each hay was sampled in quadruplicate, then, a pool of each sample was made and analyzed in duplicate with the portable NIRS.

The climate characteristics of the sampling areas are reported in Table 1.

For the polyphite hays, all the experimental areas have been represented, while for Gramineae and alfalfa hay, only three areas were sampled (Figure 2).

### 2.2. NIRS Analysis

After collection, all hay samples were analyzed using a portable NIRS (AgriNIR^TM^, Dinamica Generale, Poggio Rusco, Mantova, Italy). The devise is equipped with the NIR^TM^ Trace management software that allows you to record, organize and show analyses of forages and grains. NIRS analyses requires a sample (0.5–1.0 g) which is exposed to an electro-magnetic scan over a spectral wavelength range of 950 to 1800 nm (near infrared), each sample was analyzed in duplicate, and the mean was used for statistics. Samples are scanned in the raw state, without samples pre-treatments. Moreover, the sample box is optimized for un homogeneous samples as forages and hays. For hays, as suggested by the manufacturer, the fibers must be cut into pieces not longer than 2–3 cm, and the material is mixed, in order to have a sample as homogeneous as possible, and pressed into the sample box in order to remove as much air as possible. Energy in this spectral range is directed on to the sample and reflected energy (R) is measured by the instrument. The reflected energy is stored as the reciprocal logarithm (log 1/R) and the spectra are transformed to provide information about the chemical composition of the sample [17]. The absorbance associated with chemical bonds in the forage sample enables the identification of dry matter (DM), crude protein (CP), neutral detergent fiber (NDF), acid detergent fiber (ADF), ether extract (EE) and ash. Then, the non-fibrous carbohydrates (NFC) were calculated as follow: 100 − (NDF + CP + EE + Ash). In Table 2 the NIRS calibration parameters are reported.

### 2.3. Statistical Analysis

The data were subjected to analysis of variance according to the following equation:Yij = μ + αi + εij
where y is the experimental data; μ is the general mean; α is the production area (Avellino, Benevento, Caserta, Salerno, Piana del Sele and Vallo di Diano); and εij is the error term.

The differences among the means were compared using the Tukey test. All the statistical procedures were performed using the JMP software (JMP^®^, Version 14, SAS Institute Inc., Cary, NC, USA,1989–2021). Differences were considered statistically significant at *p* < 0.05.

## 3. Results

The chemical composition of polyphite hays according to the sampling area is reported in Table 3. No differences were detected for DM, ADF and NFC. CP content (ranging from 71.2 to 117.1 g/kg as fed) was significantly (*p* < 0.05) higher while NDF (ranging from 591.5 to 716.7 g/kg) was significantly (*p* < 0.01) lower in the hays produced in the areas of Avellino and Vallo di Diano than in those from Benevento, Caserta, Salerno and Piana del Sele. The EE was lower (*p* < 0.01) in Salerno area hays. The ash content was significantly higher in Benevento’s hays (*p* < 0.01).

The Gramineae hays (Table 4) from Piana del Sele and Vallo di Diano showed significantly higher CP (68.1 and 66.9 g/kg as fed, respectively, vs. 52.2 g/kg as fed; *p* < 0.01) and lower ADF (429.7 and 438.7 g/kg as fed, respectively vs. 467.3 g/kg as fed; *p* < 0.01) than those from Caserta. Ashes were significantly (*p* < 0.01) higher in Piana del Sele hays. No differences were observed for the others chemical parameters.

As reported in Table 5, CP content was significantly higher in alfalfa hays from Piana del Sele than in those from Vallo di Diano (175.1 vs. 128.6 g/kg as fed, *p* < 0.01) which showed a significantly (*p* < 0.01) higher ash and a lower NDF percentage compared to both the other sampling areas. No differences were detected for the other parameters.

Figure 3 shows the crude protein and NDF content of the hays evaluated in the different areas subjected to the experiment.

## 4. Discussion

A fast knowledge of the nutritional value of dairy farm forage is critical for livestock. A specialized laboratory is generally needed for quality evaluation, being rather expensive and involving a high amount of time and work. For these reasons, faster and easier tools for determining the physical and chemical characteristics of forage have been developed in recent years [18]. The Near Infrared Spectroscopy (NIRS) has been proposed as a valid alternative to this purpose since it allows to obtain fast analysis of feed nutritional quality [19,20,21].

In this research, we utilized a portable NIRS system to analyze the nutritional parameters of forages and its suitability for farmers and technicians to obtain the nutritional state of animals in farms. The most important goals were to: evaluate forage quality, increase sampling with no additional costs and obtain on-field results, thus allowing quick decisions concerning possible correctives.

In this study, the NIRS technology allowed us to analyze more than 400 samples to obtain a complete vision of the animals feeding situation in a wide area, such as the Campania Region. An adequate rationing of the animal is essential to obtain high performance, in terms of high weight gain for meat producer animals, high milk production and quality and reproductive efficiency. In this context, the quality of the forage plays an important role, being the basis for healthy feeding of ruminants. Forage quality varies considerably between and within forage crops depending on species and varieties, stage of growth, climatic conditions, soil fertilization and conservation methods [22]. When forages are harvested in the right phase of plant growth, legumes are generally more digestible [23], this happens because legumes usually have less fiber and favor higher intake than Gramineae. In fact, the stage of maturity is the most important factor in determining the quality of the forage, because it decreases as it matures [22]. Mature plants become more fibrous, the concentration of NDF increases and the intake drops dramatically. Digestibility is a parameter that varies a lot in forages, in fact, structural carbohydrates are more difficult to digest than non-fibrous components [24]; as reported by Bal et al. [25], an immature plant can be digested at 80–90%, while a mature plant at less than 50%. In order to maintain the farm’s competitiveness, to reduce feed costs and to increase the company’s protein self-sufficiency [26], it is a common practice to produce the forages on the farm, especially in southern Italy. In recent years, regions with Mediterranean-type climate have been subjected to increasing climatic stress, mainly due to the registered decrease in rainfall, as well as in the distribution of rainfall in the various months [27]; this aspect, certainly, also affects chemical characteristics of forage growth and this drier trend, indicative of climate change, is expected to continue [28]. In the Mediterranean basin, it is estimated that warming reached +4 °C in summer, resulting in a further month of summer conditions [29] and a greater shortage of water [30,31].

In this scenario, climate changes can affect crop–livestock systems, animal health and productivity, mainly acting on forage availability and quality, and, in the mixed rain-fed systems, the effects will be higher than in irrigated systems.

The possibility to cope with the effects of climate change will vary according to the area where the livestock is located, the species and breeds reared and the cultivated forages. It is known that variation in the environment may alter forage quality, even when forages are harvested at similar maturity stages [3], the main effect being on forage yield [32]. Environment influences forage quality by altering leaf/stem ratios, but it also affects plant development and the chemical composition of the different plant parts. The most important environmental factors are closely related to climate changes, regarding temperature, water availability, solar radiation, and soil nutrient availability.

Within these factors, temperature plays a critical role in determining forage quality. Optimal growth temperature ranges from 20 °C (e.g., alfalfa) 30–35 °C (e.g., corn). If temperatures are below the optimum, soluble sugars accumulate because photosynthesis is less sensitive to lower temperatures than growth. Instead, high temperatures speed up plant development and decrease leaf/stem ratios and digestibility [3]. Increasing temperature lowers forage quality; it has been reported that a +1 °C decreases digestibility of cool-season forages by 3–7 g/kg [33]. As a consequence, forages from high elevations areas, due to their lower temperatures, are generally of higher quality than those produced at low elevations.

The decrease in digestibility due to high temperatures is attributed to the higher NDF concentrations; moreover, the NDF of such forages is usually less digestible than that of forages produced at lower temperatures due to the increased lignification [32]. During spring growth, a faster decrease in forage quality occurs, due to the combination of increasing temperatures and advancing maturity [34]. On the contrary, during late-summer regrowth, temperatures are not increasing and the effect of advancing maturity often results in a slower decline in forage quality. Thus, harvesting time is critical to assure forage quality, and the opportunity for mismanagement is high.

Water stress is also a key factor to improve forage quality [32]. Water stress typically slows maturation of forages and leaf mass is reduced due to the accelerated senescence of older leaves [35]. On the other hand, if the leaf loss is not severe, water deficit may actually improve the forage digestibility.

Finally, nitrogen fertilization has a high impact on forage raising crude protein concentration in grasses. Forages with low crude protein concentration, such as warm-season grasses, can achieve an improving of digestibility after nitrogen fertilization by stimulating rumen microbe activity [32].

Climate change, particularly the global warming, may strongly affect production performances and livestock production of farm animals [36]. Heat stress is a major source of production loss in the dairy and beef industry and new knowledge about animal responses to the environment continues to be developed [37].

The results of this study confirmed these concerns and, in addition, highlight a number of historical factors and management habits that affect forage production and quality. In the Campania region, especially in the hilly areas (Avellino and Vallo di Diano), goats and sheep are mainly bred, while in the area of Piana del Sele, buffalo farms are most widespread. As reported in Table 1, the sampling areas of Avellino and Vallo di Diano are located in hilly areas, whereas Benevento, Caserta, Salerno and Piana del Sele are located in the plain. The altitude ranged from 4 to 450 m of Salerno and Vallo di Diano, respectively, whereas the temperatures ranged from mean 10°C registered in Vallo di Diano to 16.8 °C in Piana del Sele. The lower mean rainfalls were reported for Benevento and Piana del Sele areas.

The samples of polyphite hays analyzed for this study widely varied in their chemical composition as shown by the means, minimum, maximum and standard deviations. The most important factor affecting the quality of hay was the structural carbohydrate content, which mainly depends on what stage the plant is in at the time of cutting. Choosing the right moment for cutting is not always easy for the farmer, especially if there are different botanical species, such as in polyphite hays, which are characterized by different rates of growth. The weather conditions also influence the choice of the mowing period; when traditional haymaking methods are used, it is necessary to have 3–4 days of favorable time to dry the forage in the field. With the exception of alfalfa hays, polyphite and Gramineae hays appeared to be of poor quality, characterized by low protein levels and high structural carbohydrates. Due to a number of conditions (risk of rain, excessive heat whether) occurring during the cutting period (end of May–beginning of June) in the Mediterranean area under consideration, the cutting was often delayed. Plants grown at high temperatures generally produce lower quality forage than plants grown under cooler temperatures, and cool-season species grow most during the cooler months of the year [38]. This justifies the high structural carbohydrate content and the low CP levels recorded for the majority of the experimental samples, with the exception of the Avellino area and the Vallo di Diano, characterized by higher altitudes (360 and 450 m, respectively), where the temperatures are cooler compared to the other areas (13.2 and 10.0 °C for Avellino and Vallo di Diano, respectively) and a similar trend was observed for polyphite and Gramineae hays. In addition, based on a standard forage analysis, ash content can also be a sign of problems. As evidenced in Gramineae hays, in Piana del Sele the levels of ash were higher compared to the other sampling areas, probably due to the soil characteristics and/or the haymaking technique. Mowing that is too close to the ground, swathing carried out with inadequate equipment, the presence of insufficiently compact and flat soils and the collection of fodder that derives from crops with little turf, are among the most frequent causes of the problem.

According to the Dairy National Research Council [39], Gramineae and alfalfa hay typically will have 8–10 percent ash. Any more than 10 percent ash in a forage sample can be considered contamination from external sources, primarily soil added during hay harvesting or heavy rain splashing soil on to the leaves [38]. In addition, the production of Gramineae and alfalfa hays was stated only for some areas, the provinces of Caserta, Piana del Sele and Vallo di Diano, because the production of this kind of forages in the other areas is not widespread. This is probably the reason for the poor quality of the forages obtained in those areas, as well as because a large part of the production is concentrated in the hilly areas for small ruminant farms’ grazing, rather than for the production of forages for hay. Sheep and goat farming in Italy is generally characterized by a mixed production: the main one is milk, whereas meat is considered a secondary product; the consumption of lamb meat is occasional and seasonal, punctuated by religious holidays, such as Christmas and Easter [40]. Mostly, these are semi-intensive farms, thanks to the grazing of the animals, which are allocated in the stables during the coldest periods [41]. Thus, the feeding of livestock is mainly managed through grazing activities, with possible additions of feed in periods in which free access to land used for grazing is not possible or in the case in which adverse climatic conditions have affected its availability. Finally, the sedentary type farms are not very widespread, with a diet based exclusively on hay, silage and concentrates; this kind of farming, despite having a positive response in terms of management costs, is affected by the seasonality of milk production, generally concentrated in the first half of the year, with a peak in spring [42,43].

The alfalfa hay is one of the most utilized forages in Italy, especially in the areas of Po Valley involved in Parmigiano Reggiano cheese production [44]. Alfalfa is collected in several cuttings and the cuttings harvested in spring or fall have a higher leaf and protein content than summer-produced alfalfa at the same maturity, due to temperature and photoperiod [38]. In fact, the high summer temperatures increase the rate of plant maturation and cell wall lignification resulting in a reduction in digestibility. Cool temperatures retard maturity and, therefore, promote higher quality at a given age [22]. Maturity influences both fiber digestibility and protein fractions in alfalfa through increasing the leaf/stem ratio and increasing lignification of stems, which, in turn, alters fiber digestibility [45]. Alfalfa grazing requires high accuracy because young grass can be dangerous for ruminants bloating, a pathological syndrome that can even be fatal consisting of abnormal rumen swelling [46]. Haymaking is complex, especially at the first cut, when the grass is coarse due to the presence of weeds, and the season is not very favorable due to rainfall, humidity in the air and soil and poor solar radiation [44]. For these reasons, the production in Campania region is not very widespread; many farmers prefer to buy this hay only to improve the protein content of the ration.

## 5. Conclusions

This study confirms the high usefulness of the NIRS technology to obtain a fast, complete and accurate measure of forage quality. The relatively easy use of portable NIRS tools allows the nutritionist to obtain information about animal ration highlighting a number of environmental and management factors that may be critical to ensure the objective to achieve the goal of combining production, quality and animals’ health.

Our results showed a rather variable chemical composition; many factors affected forage quality and environmental condition of the sampling area, but the botanical species and the haymaking techniques also seem able to affect the hay quality. Alfalfa hay produced in Piana del Sele area seems the most promising hay, characterized by high protein levels and low structural carbohydrates. On the contrary, the polyphite and Gramineae hays produced in most of the areas of Campania region showed poor nutritional value due to the low protein content and high structural carbohydrate that significantly reduced its digestibility. The knowledge of these characteristics and the need to improve them is required by the farmers to formulate balanced rations to maintain animal health and guarantee a high level of animal products. In fact, the use in the ration of high-quality forages represents a pivotal factor to allow the production of high-quality products of animal origin.

## Figures and Tables

**Figure 1 animals-12-03035-f001:**
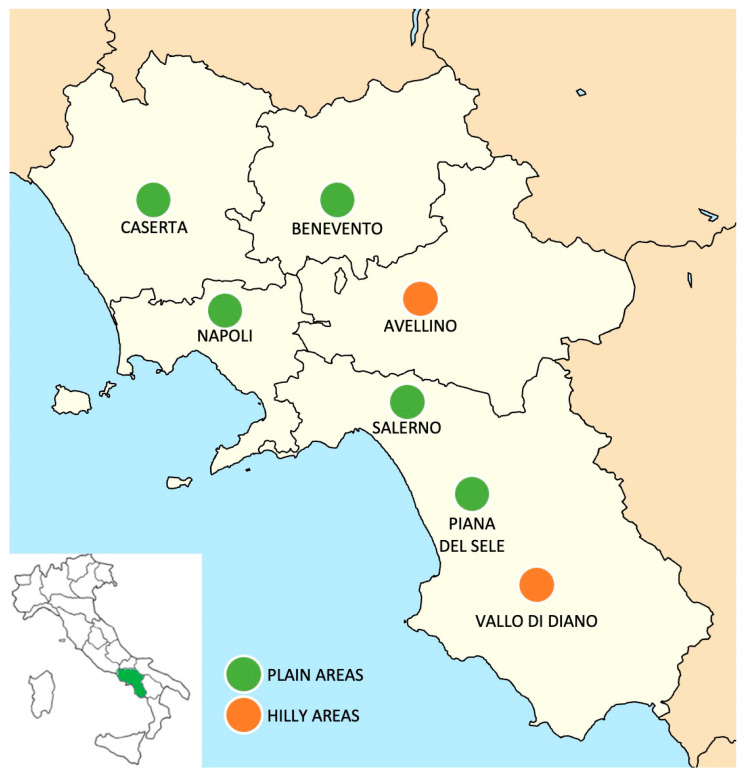
Geographical map of the different sampling sites.

**Figure 2 animals-12-03035-f002:**
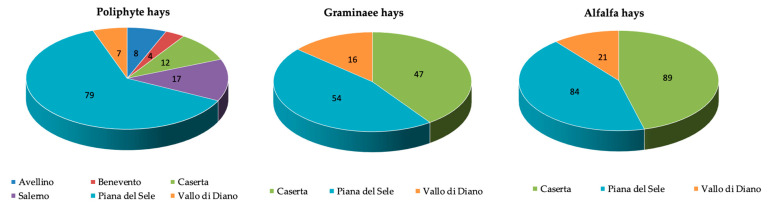
Classification of the hays according to the experimental area.

**Figure 3 animals-12-03035-f003:**
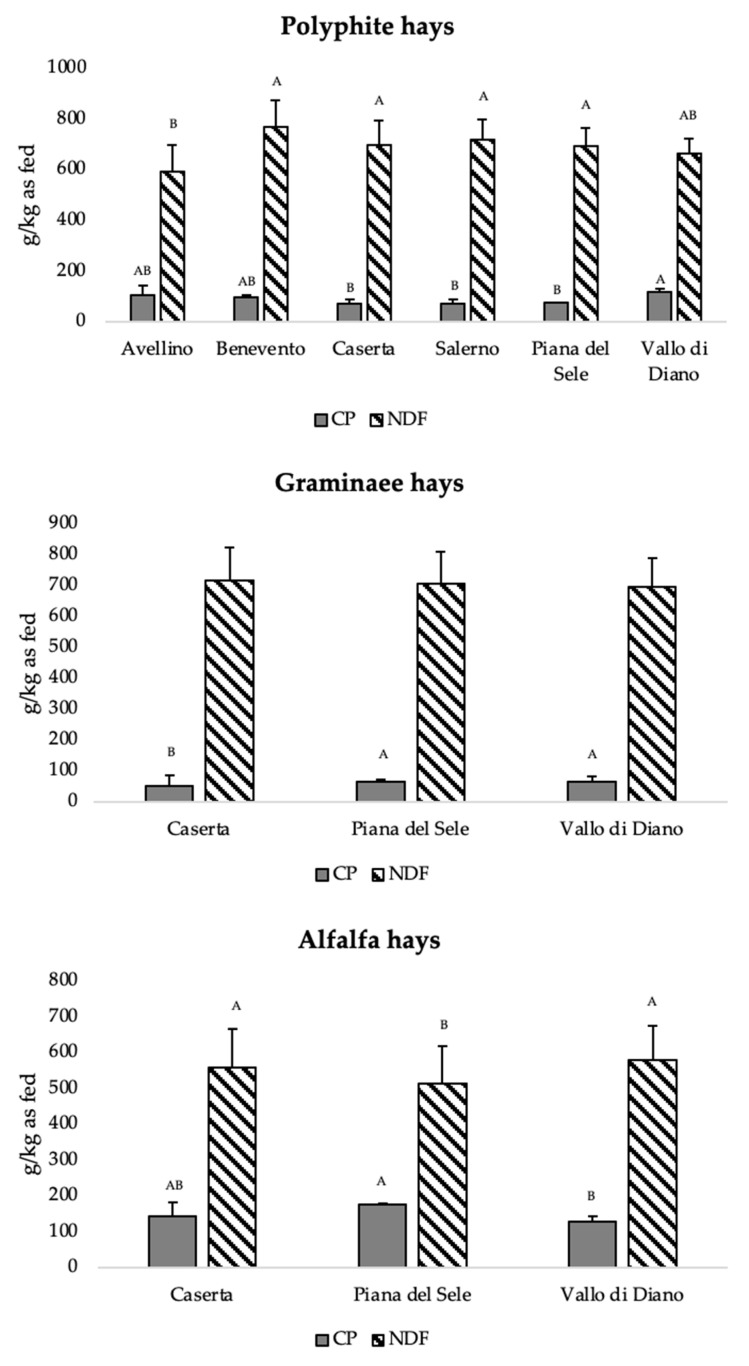
Crude protein and neutral detergent fiber content of the analyzed hays (polyphite, Gramineae and alfalfa) comparing the different sampling areas. CP, crude protein; NDF, neutral detergent fiber; A and B, different letters indicate statistically significant differences at 1%.

**Table 1 animals-12-03035-t001:** Characteristics of the sampling areas.

Area	Longitude	Latitude	Altitude *	Temperature **	Rainfall
	°E	°N	m	°C	mm
Avellino	14°47	40°55	360	13.2	1111
Benevento	14°46	41°07	154	14.4	777
Caserta	14°33	41°07	68	15.2	1153
Salerno	14°45	40°40	4	15.7	1376
Piana del Sele	14°97	40°50	10	16.8	988
Vallo di Diano	15°39	40°31	450	10.0	1192

*: a.s.l. (at sea level); **: annual mean value.

**Table 2 animals-12-03035-t002:** NIRS calibration parameters or validations of the method used for forages analysis.

Nutrient	RMSECV	R^2^	N	RMSEP	R^2^	Bias	Slope
**DM**	0.53	0.85	48	0.60	0.40	−0.21	0.45
**CP**	0.88	0.95	48	1.22	0.97	1.75	0.88
**NDF**	2.20	0.92	48	2.85	0.89	−1.83	0.84
**ADF**	1.62	0.87	48	2.11	0.80	1.03	0.80
**EE**	0.18	0.88	48	0.45	0.35	0.35	0.75
**Ash**	0.74	0.65	48	0.94	0.65	0.25	1.01

RMSECV, root means square standard error of cross validation; R^2^, standard deviation of the reference method/RMSEP; N, number of samples after outlier removal; RMSEP, root means square standard error of performance; DM, dry matter; CP, crude protein; NDF, neutral detergent fiber; ADF, acid detergent fiber; EE, ether extract.

**Table 3 animals-12-03035-t003:** Chemical composition (g/kg as fed; mean ± SD) of polyphite hays according to the sampling area.

Sampling Area	Polyphite Hays						
	DM	CP	NDF	ADF	Ash	EE	NFC
Avellino	882.6 ± 19.4	104.1 ± 36.4A	591.5 ± 106.1B	373.5 ± 56.5	100.3 ± 17.4B	15.9 ± 6.40A	188.1 ± 92.4
Benevento	883.2 ± 7.10	98.0 ± 5.0AB	766.7 ± 104.2A	448.5 ± 19.0	130.7 ± 55.7A	13.0 ± 4.07AB	134.0 ± 101.1
Caserta	890.9 ± 19.6	72.5 ± 15.9B	696.5 ± 94.1A	442.6 ± 56.3	89.0 ± 13.6B	12.3 ± 6.95AB	129.7 ± 74.5
Salerno	875.3 ± 17.6	71.2 ± 18.9B	716.7 ± 81.2A	449.5 ± 80.6	82.5 ± 14.1B	9.60 ± 2.82B	119.9 ± 96.0
Piana del Sele	879.2 ± 16.4	73.5 ± 33.8B	690.0 ± 73.5A	426.5 ± 62.3	91.3 ± 13.4B	11.7 ± 4.21AB	133.4 ± 55.5
Vallo di Diano	879.1 ± 14.2	117.1 ± 10.9A	661.2 ± 62.0AB	427.6 ± 14.0	94.3 ± 8.80B	12.7 ± 3.20AB	114.6 ± 62.0

DM, dry matter; CP, crude protein; NDF, neutral detergent fiber; ADF, acid detergent fiber; EE, ether extract; NFC, non-fibrous carbohydrates; A and B, Along the column, for each hay, different letters indicate statistically significant differences at 1%.

**Table 4 animals-12-03035-t004:** Chemical composition (g/kg as fed; mean ± SD) of Gramineae hays according to the sampling area.

Sampling Area	Gramineae Hays						
	DM	CP	NDF	ADF	Ash	EE	NFC
Caserta	891.1 ± 20.9	52.2 ± 21.0B	720.2 ± 60.9	467.3 ± 59.5A	82.5 ± 11.0B	11.2 ± 4.37	133.8 ± 57.8
Piana del Sele	889.0 ± 15.3	68.1 ± 34.4A	707.3 ± 86.1	429.7 ± 75.4B	89.2 ± 12.0A	10.5 ± 4.23	124.9 ± 74.9
Vallo di Diano	884.2 ± 11.2	66.9 ± 16.1A	698.4 ± 17.6	438.7 ± 49.4AB	79.3 ± 15.1B	10.9 ± 4.02	144.4 ± 56.5

DM, dry matter; CP, crude protein; NDF, neutral detergent fiber; ADF, acid detergent fiber; EE, ether extract; NFC, non-fibrous carbohydrates; A and B, Along the column, for each hay, different letters indicate statistically significant differences at 1%.

**Table 5 animals-12-03035-t005:** Chemical composition (g/kg as fed; mean ± SD) of alfalfa hays according to the sampling area.

Sampling Area		Alfalfa Hays					
	DM	CP	NDF	ADF	Ash	EE	NFC
Caserta	887.6 ± 17.9A	144.5 ± 38.8AB	558.5 ± 107.7A	398.8 ± 80.4	97.4 ± 15.4B	13.8 ± 5.10	185.8 ± 70.0
Piana del Sele	875.5 ± 19.5B	175.1 ± 30.0A	513.50 ± 83.5B	383.8 ± 63.6	104.3 ± 11.8A	14.4 ± 3.92	192.6 ± 60.2
Vallo di Diano	881.0 ± 8.70AB	128.6 ± 38.4B	580.61 ± 95.2A	386.8 ± 54.4	88.9 ± 11.9B	11.9 ± 3.45	190.0 ± 57.4

DM, dry matter; CP, crude protein; NDF, neutral detergent fiber; ADF, acid detergent fiber; EE, ether extract; NFC, non-fibrous carbohydrates; A and B, along the column, for each hay, different letters indicate statistically significant differences at 1%.

## Data Availability

Not applicable.

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
