# Peer review of "Nutritional Characterization of Hay Produced in Campania Region: Analysis by the near Infrared Spectroscopy (NIRS) Technology"

_animals, 2022, doi:10.3390/ani12213035_

Round 1
Reviewer 1 Report
ID: animals-1954284
Title: Nutritional characterization of hay produced in Campania region: analysis with NIRS technology
The objective of this study was to investigate the differences in hay quality in different areas of production of Campania region (Italy) by using the Near-infrared spectroscopy technology. Although the work is interesting, a few of those concern points and some questions are shown below;
Title:
-“NIRS” It should be the full word.
-What is the new finding from this study?
- Why did the author need to study the hay from those areas?
Abstract:
-L31: “(P<0.01 and P<0.05)” >>> change to (P<0.05)
- The authors should draw conclusions from this study as to what the results were and which hay was most suitable for each area too.
Introduction:
-L55-56: “(i.e. Vi-tellone bianco dell’Appennino Centrale IGP (www.vitellonebianco.it)”? Please check
-L68: “Dairy cattle breeding...” Dairy only?
-L70-71: “...improvement of animal welfare. Results of ...that improving animal welfare...” How does PLF relate to improving animal welfare?
-L82: “...avoiding time-consuming chemical analysis” The authors should demonstrate a little more about the advantages of using NIRS and the limitations of the wet analysis.
Materials and Methods:
- Why did the authors not study Graminaee hays and Alfalfa hays from the same area as Polyphite hays (Avellino, Benevento, Salerno)?
-What is the experimental design of this study? Please mention the experimental design.
- How old of each forage crop is that haymaking?
-L117: “...each sample was analyzed in duplicate...” How many replicates analyze for each sample area?
-L121-124: “The absorbance associated with chemical bonds in ... and non-structural carbohydrates (NSC).” Please specify a range of the wave number of each parameter.
-How much of the P-value that a significant effect on the treatment? Please state more clearly.
Results:
-L136: “...was significantly (P>0.01)...”? >>> change to (P<0.05)
Discussion:
-L195: “...climate changes can affect crop...” How does climate change or area differences affect nutritional value differences? / The authors should hit the point for describing the factors affecting the differences in the nutritional value of that hay as well as report on the nutritional values of previously studied hay.
-L236: “...the soil characteristics...” How do the soil characteristics affect the ash content of hay? The author should explain in a bit more detail.
-L267-268: “Lignin negatively affects forage fiber digestion .... and tissue distribution” This is a well to known matter. Also, this study did not report the effect of lignin content in each hay. Thus, the author does not need to mention it.
Conclusions:
- Hay from which area is the best or best quality to feed ruminants? /The authors should have recommendations as to what type of grass in each area is suitable for haymaking.
Table 2-4:
-“different letters indicate statistically significant differences at 1%.” It should be changed to significant differences at 5%.
Figure 3.
- It was unclear whether each graph of the parameter was statistically different. Please state the symbol more clearly.
Reviewer 2 Report
Comment to Authors
This paper highlights the importance of Infrared Spectroscopy (NIRS) technology for the qualitative characterization of hays produced in different areas of the Campania region. The paper is interesting, however it requires minor revisions.
Some points must be attended before publication:
Title Please delete with and added by
Pag 1 line 25 delete picture change with analysis
Page 5 Please describe Table 2
Page 6 line 172 Have you used the Sas or Jmp software? Please clarify
Page 7 Please improve title of Figure 3
Page 9 line 211 change picture with vision
Page 10 line 277 The sentence was be improved
Page 12 delete picture change with measure
Page 12 line 356-358 delete sentence
Page 12 line 358 delete The studied forages and added Our results
Best regards
